# Sociodemographic and Personal Predictors of Exclusive Breastfeeding in Pregnant Mexican Women Using Public Health Services

**DOI:** 10.3390/healthcare10081432

**Published:** 2022-07-30

**Authors:** Karina Serrano-Alvarado, Lilia V. Castro-Porras, Claudia I. Astudillo-García, Mario E. Rojas-Russell

**Affiliations:** 1School of Higher Studies (F.E.S.) Zaragoza, National Autonomous University of Mexico, Mexico City 09230, Mexico; kserrano@unam.mx; 2Centre for Policy, Population and Health Research, National Autonomous University of Mexico, Mexico City 04510, Mexico; liliacastrop@comunidad.unam.mx; 3Psychiatric Care Services, Health Ministry, Mexico City 10200, Mexico; claudiaiveth.astudillo@gmail.com

**Keywords:** personal predictors, exclusive breastfeeding, pregnant, Mexican women, public health services, predictor model

## Abstract

Exclusive breastfeeding (EBF) is a cost-effective healthy behavior for the mother–child dyad. Globally, rates of EBF are low. Little research has been conducted on the joint role of modifiable and nonmodifiable variables in pregnant women’s decision-making. The aim was to develop and test a model that used personal and sociodemographic factors to predict whether pregnant women who use public healthcare services plan to breastfeed. In a nonprobabilistic sample of 728 pregnant Mexican women, self-efficacy, the planned behavior theory constructs, and the intention to breastfeed (BFI) were measured. A total 60% of the sample was randomly chosen to develop a predictive multivariate logistic regression model. The model was validated in the remaining 40%. Women in the highest tertiles of attitudes and self-efficacy had fourfold increased chances of having a high BFI (OR 4.2, 95% CI [2.4, 7.4]). Working was associated with a decreased intention to exclusively breastfeed (OR 0.61, 95% CI [0.37, 0.98]). The model predicted BFI with a sensitivity of 38.3% and specificity of 82.2%. While personal variables predict the BFI, working decreases women’s chances of breastfeeding. The results can be utilized to develop primary prevention strategies to help mothers who use public health services to breastfeed.

## 1. Introduction

Exclusive breastfeeding (EBF) is a type of infant feeding in which the baby receives only breast milk and no other liquids or solid foods, except for vitamins and medications. The World Health Organization (WHO) and the United Nations Children’s Fund (UNICEF) recommend using EBF for the first six months of life, followed by introducing age-appropriate and safe foods in conjunction with breastfeeding (BF) for up to two years or longer [1].

Although BF is associated with multiple health and economic benefits for mothers and children [2,3], the recommended breastfeeding rate is low worldwide. The World Health Organization’s Global Nutrition Target (WHO GNT) proposes to achieve ≥70% LME compliance by 2030 [4]. Bhattacharjee et al. [5], in a recent study on the prevalence of exclusive breastfeeding in 94 low- and middle-income countries (LMICs), found that in 2018, roughly 55.2% of children under six months of age were not exclusively breastfed in the countries studied, which represents a reduction of only 4.8% since the year 2000.

Despite recommendations that breastfeeding should begin immediately after birth, in Mexico, data from the 2018–19 National Health and Nutrition Survey (ENSANUT) indicate that 47.7% of infants are breastfed within the first hour of life [6]. Fewer than a third of six-month-old children are exclusively breastfed (28.6%), while nearly half of children between 12 and 15 months (46.9%) and almost a third of children between 20 and 23 months (29%) continue to receive breast milk [6,7]. In spite of recent increases in BF rates, Mexico remains far from achieving the WHO recommendations for breastfeeding practices [8], so to protect, promote, and encourage breastfeeding, research, and predictor identification is greatly relevant.

Women face different structural, environmental, and individual determinants that affect the decision to breastfeed, from public prenatal care policies to aspects related to their immediate family, personal, and health environment [9].

One of the most significant sociodemographic barriers to starting or continuing BF among working women is the return to work [10,11]. Labor spaces in low-income countries do not provide adequate conditions for combining paid work with BF practice [12].

Regarding personal variables, EBF has been identified as a health-related behavior [13], and behavioral intention has been one of the most studied constructs in relation to health behaviors.

Intention is the willingness to initiate, modify, or change a behavior. It starts from the assumption that intentional behavior is goal-oriented; so, it can be explained by the change processes that occur during the decision-making process, including cognitive evaluations and behavioral resources that are required to take or sustain action [14]. Various reviews, including meta-analyses, show that intention predicts BF and its duration [15,16,17,18], and is also one of the modifiable variables with the greatest impact on lactation. A higher intention to perform BF and a higher likelihood of practicing BF after childbirth are associated with other Theory of Planned Behavior (TBP) variables such as attitudes, subjective norms, perceived control, and self-efficacy [16,19,20]. Women who intend to exclusively breastfeed are more likely to achieve their goals. Compared with women who do not intend to breastfeed exclusively, those who do are roughly twice as likely to achieve their goals [21,22]. Other reports have found that mothers who want to breastfeed their babies prenatally are more likely to breastfeed them and use infant formula less frequently in the early postpartum period [23]. Moreover, there is evidence that the intention to become pregnant is associated with a longer duration of breastfeeding [24]; when women and their partners plan when to have children, they anticipate decisions regarding infant feeding.

Self-efficacy [25] is another construct that has been demonstrated to be a good predictor of BF. Past breastfeeding experiences, observational learning from competent role models, verbal persuasion from trusted individuals, and maternal emotional calm linked with nursing help increase the likelihood of breastfeeding initiation and maintenance [26].

Furthermore, a woman’s economic conditions may influence her decision-making. According to Hmone et al., [17], in a cross-sectional study of 353 pregnant women at 28–34 weeks’ gestation, women who worked had a reduced intention to exclusively breastfeed (adjusted odds ratio (AOR) = 0.30; 95% confidence interval (CI) 0.17–0.53). Women from wealthy (AOR = 2.43, CI 1.08–5.47) and middle-income (AOR = 1.79, CI 1.01–3.16) families, those with high (AOR = 10.19, CI 3.43–30.23) and middle (AOR = 5.46, CI 1.79–16.72) knowledge about BF, and those who received information from health professionals (AOR = 2.29, CI 1.29–4.03) and mobile devices (AOR 3.62, CI 2.0) had a higher intention to breastfeed exclusively.

As a result, while the TPB components are substantial predictors of intention to breastfeed, decision-making is also influenced by sociodemographic characteristics, which could explain why the predictive ability of intention to breastfeed is inconsistent. Most of the research has focused on one form of determinant or the other separately, with relatively little research combining the two types of determinants in the same study. Thus, this study aimed to identify, from a set of personal, sociodemographic, and pregnancy-related variables, those that best predicted the intention to exclusively breastfeed in a sample of pregnant women who used public health services at the first level of care.

## 2. Materials and Methods

### 2.1. Design

The predictive validity of a multivariate logistic regression model for the intention to breastfeed was tested in a cross-sectional study. The sample was randomly divided into two groups, with 60% of the participants in the first group and 40% in the second. The logistic model was developed in the first subsample, and its predictive capacity was confirmed in the second one.

Multivariate logistic regression allows the simultaneous effect of multiple factors on a dichotomous outcome [27]. It can be employed as a predictive analysis strategy by using existing data to predict outcomes that are not available.

The study, including ethical considerations, was approved by the funding institution with number IN307916-IT300621.

### 2.2. Setting

Between November 2017 and June 2018, pregnant women from two different health institutions in the states of Aguascalientes and Mexico were invited to participate in this study, which focuses on breastfeeding intention.

In the country’s central zone, to which the states of Aguascalientes and Mexico City belong, only 22.7% of infants under six months received exclusive breastfeeding [7], compared with 28.6%, which represents the national average [6].

### 2.3. Sample

The study used a convenience sample with the following inclusion criteria: being pregnant at the interview, being able to read and write, and using one of the primary care centers where the study was conducted.

In total, 728 participants were formally enrolled in the study (Figure 1). Those with high-risk pregnancies or any disease that makes breastfeeding impossible were excluded. Due to missing data in the information obtained, 10 participants in the development and 5 in the validation samples were eliminated. The remaining 713 women were considered in the study. All volunteers gave written informed consent before being enrolled in the study.

### 2.4. Measurement

A researcher-applied questionnaire was designed to collect data on demographic characteristics, pregnancy characteristics, and the variables of the extended version of the PBT: intention to breastfeed, attitudes, subjective norm, perceived behavioral control, affect anticipation, and breastfeeding self-efficacy. A comprehensive description of the psychometric characteristics of the instruments administered can be found elsewhere [28]. Standardized interviewers administered the questionnaires in the health center waiting rooms.

The sociodemographic variables assessed are as follows: age, categorized in 3 groups—less than 21 years = 0, 22–27 years = 1, and 28 years or more = 2; the perceived socioeconomic level was assessed with a 10-point analogue visual scale, for comparison purposes it was coded in low = 0 and high = 1 considering the median (5 points) as cut-off; the educational level was determined from the report of the last grade of school completed and was classified as middle or less = 0 and high school or more = 1; marriage status was categorized as single = 1 or married/consensual union = 2; labor condition was not working = 0 if they had no formal or informal paid employment or working = 1 if they did.

Three variables related to pregnancy were considered: planned pregnancy, referring to the planned decision to become pregnant, divided into unplanned = 0 and planned = 1; gestational age at the time of the interview, evaluated by the number of weeks and days of pregnancy reported; and parity, defined as the number of previous pregnancies and classified as primiparous = 0 for women with a single current pregnancy and multiparous = 1 for those with multiple pregnancies.

The outcome variable was the intention to exclusively breastfeed (EBI); women were asked how confident they felt about EBF their child from birth to six months, and responses were scored on an 11-point scale (from not at all confident to totally confident).

Attitudes were defined as the personal disposition to respond favorably or unfavorably toward breastfeeding [29]. They were assessed with a semantic differential scale of 16 pairs of opposing adjectives.

The subjective norm corresponds to normative beliefs about breastfeeding and the importance attributed to the opinion of breastfeeding [30]; for this purpose, 12 Likert-type items grouped into two factors that explain 90% of the variance were evaluated.

Perceived behavioral control of breastfeeding was defined as the mother’s judgment of the degree to which she can successfully implement the breastfeeding [31]. A scale was used that evaluates situations that may facilitate or hinder the practice of breastfeeding, containing 24 Likert-type items ranging from completely disagree to completely agree, grouped into three factors that explain 81% of the variance.

Affect anticipation was defined as emotional expectations about the consequences of performing a behavior [32]. The scale is composed of 13 items and two factors that explain 91% of the total variance of the instrument.

In addition, self-efficacy was defined as the mother’s confidence in her ability to breastfeed her baby successfully. For this purpose, the Breastfeeding Self-Efficacy Scale was used, including 15 Likert-type items with four response options and values from 0 to 3 [33].

### 2.5. Data Analysis

The sample was divided into two random groups; the first considered 60% of the total participants and was used as the development sample and the remaining was used as the validation sample. Descriptive statistics were used to characterize the participants’ sociodemographic characteristics and were reported as means and standard deviations. Differences between the two samples were tested using the χ2 test for qualitative variables and the Mann–Whitney test for continuous variables. Intention to breastfeed was classified into low and high based on the distribution observed in the scale scores, and logistic regression was used to develop the model for estimating it.

All sociodemographic and pregnancy-related variables were considered of interest and entered into the initial model. The variables in the PBT model, including self-efficacy, were categorized into tertiles to identify differences between extreme high and low scores. Two dimensions of the subjective norm scale (mother-in-law opinion and sister-in-law opinion) and the affect anticipation scores (positive emotions and negative emotions), however, were only divided into high and low scores because of how the response values were distributed.

Due to the high skewness of its distribution, EBI was categorized as low = 0 for women who scored their intention to breastfeed between 0 and 9 points and high = 1 for those who scored 10 (completely confident) in bivariate and multivariate logistic models.

This process obtained several models; the most parsimonious model using the forward method was chosen. ORs were estimated with 95% confidence intervals for the selected predictors. Goodness of fit was assessed from the Hosmer–Lemeshow test. Finally, we validated the model on a subsample, the technique most used in studies aimed at developing predictive models. Sensitivity, specificity, and predictive values of intention to breastfeed were also calculated. The probability of breastfeeding from the proposed model was shown through a line graph. Statistical analyses were performed with the statistical software Stata® 14 (StataCorp LLC, College Station, TX, USA).

## 3. Results

### 3.1. Sociodemographic Data

For comparison purposes, Table 1 shows the homogeneity of the personal and pregnancy-related characteristics for the development and validation samples. No statistically significant differences were found in sociodemographic characteristics and pregnancy distribution across samples.

Table 2 describes the sociodemographic characteristics of the development sample. Sixty-three percent of the participants reported a high intention to exclusively breastfeed. Approximately 80% of participants were married or living with a partner, the median age was 24 years (IQR (Interquartile range) 20 and 29 years), 50% had secondary education or less, and 57% were of low socioeconomic status. On average, women were interviewed at 27 weeks of gestation (IQR 20–33 weeks). About 60% reported having at least one child, and slightly less than half indicated not having planned their current pregnancy.

Statistically significant differences were identified between the levels of intention by age group and employment status, with younger, nonworking participants reporting a lower intention to breastfeed. There were no differences in intention to breastfeed by socioeconomic level, education, marital status, or pregnancy-related variables (Table 2).

### 3.2. Bivariate Results

In the case of the psychological variables (Table 3), statistically significant differences by level of intention were found in the expected theoretical meaning. Higher scores on the attitude dimensions were linked to a high intention to breastfeed exclusively, implying that perceiving breastfeeding as pleasant, comfortable, satisfactory, or adequate/correct (positive evaluation of the symptoms, emotions, and naturalness of breastfeeding) was linked with a higher willingness to breastfeed exclusively. Lower intention was associated with greater perceived difficulties, such as perceived stress or a lack of time, as well as perceived difficulty breastfeeding in public (perceived behavioral control). Perceived control over situations that promote calm and relaxed moods during breastfeeding, on the other hand, was related to a greater intention to exclusively breastfeed. Regarding the subjective norm, the mother of the participant woman and the doctor’s positive opinion on breastfeeding were related to a high intention to breastfeed. Self-efficacy was also associated with a high intention to breastfeed. Anticipation of positive emotions related to breastfeeding was associated with a low intention to breastfeed, contrary to expectations.

In bivariate regression analysis (Table 4) for intention to breastfeed, statistically significant associations were obtained in the age group of 22–27 years and between employment status. Regarding pregnancy-related variables, only gestational week showed marginal associations with intention (OR 1.02, 95% CI 0.99–1.04, *p* < 0.10). Most of the personal variables were associated in the expected direction with intention; in the case of the subjective norm, only the mother’s and doctor’s opinions were associated with intention to breastfeed.

### 3.3. Multivariate Results

The final multivariate model is shown in Table 5 with the best predictors identified (Hosmer–Lemeshow goodness-of-fit test, *p* = 0.77). In terms of personal variables, it was found that those in the highest tertiles were more likely to have the greatest intention to exclusively breastfeed. Women with positive attitudes towards the emotions experienced during breastfeeding were up to four times more likely to have a high intention to breastfeed, OR = 4.19 (95% CI 2.39, 7.37). Only the highest tertiles of self-efficacy (OR = 2.12 (95% CI 1.20, 3.72)) and subjective norm (physician opinion) (OR = 2.04 (95% CI 1.25, 3.32)) positively predicted breastfeeding intention.

Among the sociodemographic variables, women in the 22–27 age group were twice as likely to have a high intention to breastfeed compared with those in the other age groups. On the other hand, work negatively predicted intention to breastfeed; women who reported working were almost 40% less likely to intend to breastfeed exclusively.

Pregnancy planning, parity, and weeks of gestation at the time of the interview were not statistically significant predictors, so they were not included in the final model.

The final model showed 38.3% and 82.2% sensitivity and specificity, respectively, when tested on the validation sample; in addition, the model’s goodness-of-fit tested on the validation sample was adequate (Hosmer–Lemeshow goodness-of-fit test, *p* = 0.74).

The marginal effects of the model variables according to employment status were tested since, among the predictive variables of the model, work was the nonmodifiable factor that showed the highest predictive value.

Figure 2 shows the predicted probabilities of intention to breastfeed. In general, the observed trend indicates that regardless of employment status, being in the highest tertile of the attitude and self-efficacy scales predicts a higher probability of having a high intention to breastfeed. According to the attitude scale, a woman who does not work is about 80% more likely to intend to breastfeed than a woman who does work (67%) in the same tertile. In the case of self-efficacy, the probability of high intention is 70% for nonworking women in the highest tertile and 56% for working women in the same group.

The subjective norm (doctor’s opinion) significantly predicted the intention to breastfeed. However, contrary to the expected trend, women in the second tertile were more likely to have a higher intention to breastfeed than women in the highest tertile. On the other hand, among working women, the probability was 74% and 60% for nonworking women.

The trend also indicates that the higher the age, the higher the probability of having a high intention to breastfeed regardless of employment status (69% for nonworking women in the 28+ age group and 55% for working women in the same age group).

We observed no interaction between the model variables; the marginal effects were not statistically significant by employment status or between the tertiles of the scales, except for the attitude scale only for women who do not work.

## 4. Discussion

The goal of this study was to identify a model to predict the intention to breastfeed exclusively based on a set of personal, sociodemographic, and pregnancy-related variables in a sample of pregnant women who used public health services at the first level of care.

Positive emotional attitudes toward breastfeeding, more confidence in their personal ability to breastfeed, and a positive view of the doctor’s opinion on breastfeeding were all significant predictors of the intention to exclusively breastfeed. On the other hand, maternal work was the most important factor associated with a low intention to breastfeed, reducing intention by 40%.

Several studies have independently documented the relationship between intention to breastfeed and Theory of Planned Behavior variables [17,34,35], sociodemographic variables [36,37], or biophysical or pregnancy-related variables [38]. However, there is limited information on the combined effect of all these characteristics on pregnant women’s intention to exclusively breastfeed in primary care settings, which is an important strength of our study.

Psychological (personal) variables such as attitudes and beliefs are defined as modifiable factors. In contrast, sociodemographic and biological parameters such as parity, gestational age, and so on are nonmodifiable variables. When the focus is on the interaction of personal variables on intention to breastfeed and breastfeeding, they are typically investigated independently. According to review articles [15,16,18], the nonmodifiable variables are used as adjustment variables to standardize comparison conditions, making it difficult to understand how specific individual factors affect health behaviors [13].

In this study, the prediction model found that the mother’s employment status was the most important nonmodifiable variable. Therefore, we were particularly interested in figuring out the predictive role of modifiable personal variables when this variable is present. Postestimation analyses showed a clear trend that, even among working women, the higher the intention to breastfeed, the stronger the personal beliefs about breastfeeding.

Maternal work has been associated with an increased risk of breastfeeding cessation at 6 months of age, while modifiable factors, such as attitudes, breastfeeding knowledge, and intention to breastfeed, decrease the risk of early cessation [39,40]. On the other hand, breastfeeding-supportive workplaces have a positive effect on perceived breastfeeding self-efficacy and are associated with longer breastfeeding duration [41].

Incorporating breastfeeding policies in the workplace has benefits both for the mother–child dyads and for the companies themselves, including greater intention to breastfeed and longer duration of breastfeeding, lower work absenteeism for health reasons, higher labor productivity, and a better image of the companies [11]. Therefore, our results can be a useful tool to identify and enhance skills and personal beliefs to promote the intention to breastfeed in working women, both in primary care centers and in workplaces that support breastfeeding.

The data about the relationship between subjective norm and intention to breastfeed confirm that physician opinion has a positive effect on the intention to breastfeed, even when this opinion is only moderately valued. As Hmone et al. [17] found, breastfeeding information from health professionals is a significant predictor of the intention to exclusively breastfeed (AOR = 2.29, CI = 1.29–4.03). Further presented are differences between working and nonworking women.

Compared with younger women and women over the age of 28, maternal ages between 22 and 27 years better predicted intention to breastfeed in this study sample. In a prospective study of 6443 nulliparous women by Baumgartner et al. [42] discovered that the higher the maternal age, the higher the intention to breastfeed. However, according to other studies [35], the mother’s age is not a significant predictor of any of the outcome variables. When the effects of age are analyzed with the effects of other variables, this shows that age may be a predictor of intention.

The low variability in the results on the personal assessment scales, which indicates a favorable bias, is one of the study’s limitations. Overall, pregnant women express a strong desire to breastfeed, despite the fact that only a small percentage of them really do [43], which could explain the high scores on the scales that the PBT uses to predict intention. Another issue that could skew the results was the way the data were gathered. Even though participants were told their replies would not alter the prenatal treatment they received, the collection of information in the waiting rooms of healthcare institutions may have enhanced the probability of social desirability bias.

Another limitation was the prediction model’s sensibility, which implies that the model has a high probability of failing to appropriately identify women with a strong desire to breastfeed. However, the model’s specificity (82.3%) provides for more precise identification of women who have a low intention to breastfeed, which is especially beneficial for identifying women in the highest risk category because it can lead to decreased breastfeeding involvement or early discontinuation.

The way the data were analyzed was the study’s main strength. The predictive model was built with one random subsample and validated with another. This gives the results more credibility.

## 5. Conclusions

A prediction model for the intention to exclusively breastfeed was developed and validated in this study. The model had a good predictive ability. Even in the presence of unfavorable conditions for breastfeeding, such as maternal work, modifiable personal characteristics such as attitudes, self-efficacy, and subjective norms appear to have significant predictive value.

These findings enable the identification of modifiable personal characteristics to develop timely intervention strategies that may improve breastfeeding possibilities for women who seek care from first-contact healthcare providers.

## Figures and Tables

**Figure 1 healthcare-10-01432-f001:**
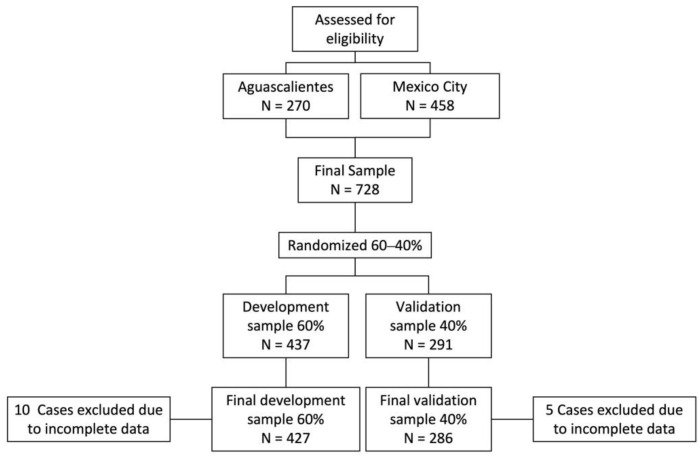
Sample selection.

**Figure 2 healthcare-10-01432-f002:**
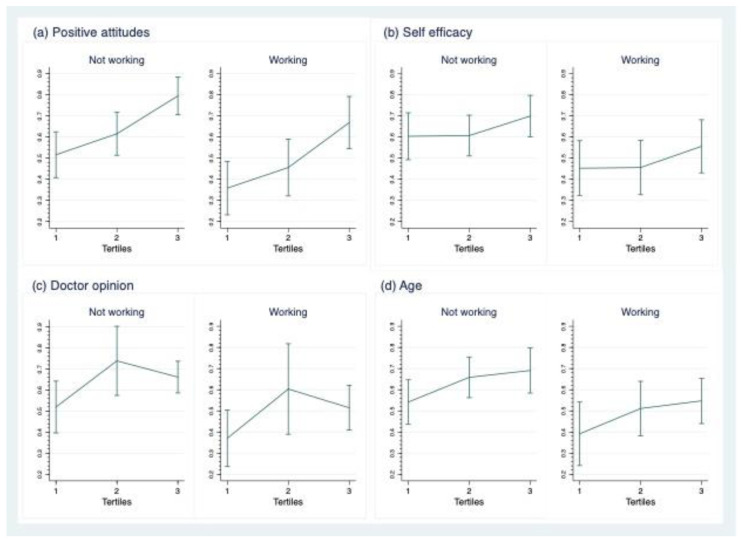
Predicted probabilities of exclusive breastfeeding intention (EBI) with 95% confidence interval by working status (WS). (**a**) Predicted probabilities of EBI at tertiles of positive attitudes scale by WS. (**b**) Predicted probabilities of EBI at tertiles of self-efficacy scale by WS. (**c**) Predicted probabilities of EBI at tertiles of doctor opinion scale by WS. (**d**) Predicted probabilities of EBI at tertiles of age group by WS.

**Table 1 healthcare-10-01432-t001:** Distribution of sociodemographic and pregnancy characteristics of development and validation samples ^a^.

Characteristics	Development Sample		Validation Sample	
	%	[95% CI]	%	[95% CI]
N	427		286	
**Age (years)**				
≤1	34.4	[30.0, 39.0]	29.0	[24.0, 34.5]
22–27	31.3	[27.1, 35.9]	34.6	[29.3, 40.3]
≥8	34.1	[29.8, 38.8]	36.4	[30.9, 42.1]
**Socioeconomic level**				
Low	57.1	[52.4, 61.8]	54.5	[48.7, 60.3]
High	42.8	[38.2, 47.6]	45.4	[39.7, 51.3]
**Marital status**				
Single	23	[19.2, 27.2]	19.6	[15.4, 24.6]
Married/Consensual union	76.7	[72.7, 80.8]	80.4	[75.4, 84.6]
**Working status**				
Not Working	66.5	[61.9, 70.8]	64.6	[58.9, 70.0]
Working	33.5	[29.1, 38.1]	35.3	[29.9, 41.1]
**Education level**				
Middle school or less	49.9	[45.1, 54.6]	51.7	[45.9, 57.5]
High school or more	50.1	[45.4, 54.8]	48.2	[42.4, 54.1]
**Planned pregnancy**				
No	46.6	[41.8, 51.4]	46.8	[41.1, 52.7]
Yes	53.4	[48.6, 58.1]	53.1	[47.3, 58.9]
Weeks of pregnancy #	27	[20, 33]	26.0	[20, 32]
**Parity**				
Primiparous	43.7	[39.0, 48.4]	41.9	[36.3, 47.8]
Multiparous	56.3	[51.6, 60.9]	58.0	[52.2, 63.7]

^a^ Differences estimated by Chi2 test for categorical variables and U Mann–Whitney tests for continuous variables; # Median [Quartile Q1, Q3]; Abbreviations: CI, confidence interval.

**Table 2 healthcare-10-01432-t002:** Sample characteristics stratified by level of breastfeeding intention of development sample ^a^.

Characteristics	Total		Low Intention		High Intention		
	%	[95% CI]	%	[95% CI]	%	[95% CI]	
Sample size n (%)	427		156 [0.36]		271 [0.63]		
**Age**							*
<21 years	34.4	[30.0, 39.0]	42.3	[34.7, 50.3]	29.8	[24.7, 35.6]	
22–27	31.3	[27.1, 35.9]	21	[15.4, 28.3]	37.3	[31.7, 43.2]	
≥28	34.1	[29.8, 38.8]	37	[29.3, 44.4]	32.8	[27.5, 38.7]	
**Socioeconomic level**							
Low	57.1	[52.4, 61.8]	55	[47.2, 62.8]	48.3	[39.3, 57.4]	
High	42.8	[38.2, 47.6]	45	[37.2, 52.8]	51.7	[42.6, 60.7]	
**Marital status**							
Single	23	[19.2, 27.2]	25	[18.8, 32.5]	21.8	[17.3, 27.2]	
Married/Consensual union	76.7	[72.7, 80.8]	75	[67.5, 81.2]	78.1	[72.8, 82.7]	
**Working status**							*
Not Working	66.5	[61.9, 70.8]	60	[52.3, 67.7]	70.1	[64.3, 75.3]	
Working	33.5	[29.1, 38.1]	40	[32.3, 47.7]	29.9	[24.7, 35.6]	
**Education level**							
Middle school or less	49.9	[45.1, 54.6]	52	[44.0, 59.7]	48.7	[42.8, 54.7]	
High school or more	50.1	[45.4, 54.8]	48	[40.3, 55.9]	51.3	[45.3, 57.2]	
**Planned pregnancy**							
No	46.6	[41.8, 51.4]	49	[41.5, 57.2]	45	[39.1, 51.0]	
Yes	53.4	[48.6, 58.1]	51	[42.7, 58.5]	54.9	[48.9, 60.8]	
Weeks of pregnancy #	27	[20, 33]	26	[18, 32]	27	[20, 33]	
**Parity**							
Primiparous	43.7	[39.0, 48.4]	47	[39.0, 54.7]	41.8	[36.1, 47.8]	
Multiparous	56.3	[51.6, 60.9]	53	[45.3, 60.9]	58.1	[52.1, 63.9]	

^a^ Differences estimated by Chi2 test for categorical variables and U Mann–Whitney tests for continuous variables; * *p* < 0.05; # Median [Quartile Q1, Q3]; Abbreviations: CI, confidence interval.

**Table 3 healthcare-10-01432-t003:** Psychological variables by level of breastfeeding intention ^a^.

Personal Characteristics	Total		Low Intention		High Intention		
	Median	[Q1, Q3]	Median	[Q1, Q3]	Median	[Q1, Q3]	
Sample size n (%)	427		156 [36.5]		271 [63.5]		
**Attitudes**							
Positive attitudes to physical symptoms	21	[15, 27]	20	[15, 24]	23	[16, 28]	*
Positive attitudes to emotions	64	[57, 70]	61	[48, 66]	66	[60, 70]	*
Positive attitudes to naturalness breastfeeding	39	[34, 40]	37	[32, 40]	40	[35, 40]	*
**Perceived Behavioral Control**							
Perceived difficulty to breastfeed in public	7	[5, 9]	7	[5, 10]	6	[4, 9]	*
Perceived of tension and lack of time	10	[8, 13]	11	[9, 14]	10	[7, 13]	*
Perceived of comfort and calm	27	[22, 31]	26	[22, 30]	28	[23, 31]	*
**Subjective Norm**							
Family opinion	3	[1, 9]	3	[1, 9]	3	[0, 9]	
Partner opinion	6	[3, 9]	6	[3, 9]	9	[3, 9]	
Mother opinion	4	[1, 9]	4	[0, 9]	6	[2, 9]	*
Sister(s)-in-law opinion	1	[0, 4]	1	[0, 4]	1	[0, 6]	
Mother-in-law opinion	2	[0, 9]	2	[0, 6]	3	[0, 9]	
Doctor opinion	9	[4, 9]	6	[4, 9]	9	[6, 9]	*
Anticipated feelings							
Positive feelings	1	[0, 4]	2	[0, 4]	0	[0, 4]	*
Negative feelings	4	[0, 9]	6	[0, 9]	3	[0, 9]	*
Breastfeeding self-efficacy	37	[28, 45]	33	[24, 42]	38	[29, 46]	*

^a^ Median differences estimated by U Mann–Whitney test; * *p* < 0.05.

**Table 4 healthcare-10-01432-t004:** Logistic bivariate associations by breastfeeding intention ^a^.

Variable	n	OR	[95% CI]	
**Sociodemographic variables**				
**Age. Reference: ≤21 years**	427			
22–27		2.49	[1.49, 4.15]	*
≥28		1.27	[0.79, 2.02]	
**Socioeconomic level. Reference: Low**	427			
High		0.87	[0.59, 1.31]	
Marital Status. Reference: Single	426			
Married/consensual union		1.19	[0.75, 1.89]	
**Working status. Reference: Not working**	427			
Working		0.64	[0.43, 0.98]	*
**Education level. Reference: Middle school or less**	427			
High school or more		1.14	[0.77, 1.69]	
**Pregnancy variables**				
**Planned pregnancy. Reference: No**	427			
Yes		1.19	[0.80, 1.77]	
**Parity. Reference: Primiparous**	426			
Multiparous		1.22	[0.82, 1.81]	
**Weeks of pregnancy**	426	1.02	[0.99, 1.04]	+
**Personal variables by tertiles**				
**Attitudes**				
Positive attitudes to physical symptoms. Reference: tertile 1	427			
tertile 2		0.95	[0.59, 1.53]	
tertile 3		2.32	[1.39, 3.89]	*
Positive attitudes to emotions. Reference: 1 tertile 1	427			
tertile 2		2.46	[1.51, 4.00]	*
tertile 3		5.11	[3.02, 8.65]	*
Positive attitudes to naturalness breastfeeding. Reference: tertile 1	427			
tertile 2		1.23	[0.72, 2.09]	
tertile 3		2.43	[1.53, 3.85]	*
**Perceived Behavioral Control**				
Perceived difficulty to breastfeed in public. Reference: tertile 1	426			
tertile 2		0.79	[0.46, 1.37]	
tertile 3		0.57	[0.34, 0.97]	*
Perceived of tension and lack of time. Reference: tertile 1	426			
tertile 2		0.63	[0.36, 1.09]	+
tertile 3		0.45	[0.27, 0.78]	*
Perceived of comfort and calm. Reference: tertile 1	424			
tertile 2		1.12	[0.69, 1.82]	
tertile 3		1.75	[1.06, 2.88]	*
**Subjective Norm**				
Family opinion. Reference: tertile 1	413			
tertile 2		0.74	[0.44, 1.25]	
tertile 3		0.88	[0.55, 1.42]	
Partner opinion. Reference: tertile 1	403			
tertile 2		0.9	[0.50, 1.62]	
tertile 3		1.15	[0.73, 1.82]	
Mother opinion. Reference: tertile 1	415			
tertile 2		1.72	[1.04, 2.84]	*
tertile 3		1.88	[1.16, 3.05]	*
Sister(s)-in-law opinion. Reference: low #	358			
High		1.00	[0.65, 1.54]	
Mother-in-law opinion. Reference: low #	377			
High		1.24	[0.81, 1.88]	
Doctor opinion. Reference: tertile 1	419			
tertile 2		1.22	[0.65, 2.29]	
tertile 3		2.41	[1.54, 3.79]	*
**Anticipated feelings**				
Positive feelings. Reference: Low #	427			
High		0.51	[0.26, 1.03]	+
Negative feelings. Reference: Low #	425			
High		0.78	[0.52, 1.16]	
**Self-efficacy**				
Breastfeeding self-efficacy. Reference: tertile 1	427			
tertile 2		1.16	[0.72, 1.87]	
tertile 3		2.85	[1.71, 4.73]	*

^a^ Outcome: low vs. high breastfeeding intention; * *p* < 0.05; + *p* < 0.10; Abbreviations: OR, odds ratio; CI, confidence interval. # The scales were split into high and low scores due to the distribution of response values.

**Table 5 healthcare-10-01432-t005:** Logistic multivariate model for breastfeeding intention ^a^.

Variable	OR	[95% CI]	
**Positive attitudes to emotions: Reference tertile 1**			
Tertile 2	2.43	[1.44, 4.07]	*
Tertile 3	4.19	[2.39, 7.37]	*
**Breastfeeding self-efficacy: Reference tertile 1**			
Tertile 2	0.89	[0.53, 1.51]	
Tertile 3	2.12	[1.20, 3.72]	*
**Doctor opinion: Reference tertile 1**			
Tertile 2	1.16	[0.58, 2.29]	
Tertile 3	2.04	[1.25, 3.32]	*
**Age: Reference < 21 years**			
22–27	2.42	[1.37, 4.26]	*
≥ 28	1.38	[0.79, 2.39]	
Working status: Reference Not working			
Working	0.61	[0.37, 0.98]	*

^a^ Outcome: low vs. high breastfeeding intention; * *p* < 0.05; Abbreviations: OR, odds ratio; 95% CI, 95% confidence interval.

## Data Availability

The data sets used and/or analyzed during the current study are available from the corresponding author on reasonable request.

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
