# Peer review of "Sociodemographic and Personal Predictors of Exclusive Breastfeeding in Pregnant Mexican Women Using Public Health Services"

_healthcare, 2022, doi:10.3390/healthcare10081432_

Round 1

Reviewer 1 Report

I suggest to review the following paragraph: the recommended breastfeeding rate is low worldwide. , in a recent study on the prevalence of extended breastfeeding in 94 low- and middle-income countries (LMICs), found that in 2018, roughly 55.2% of children under six months of age were not exclusively breastfed in the countries studied. In Mexico, the
figure was 760,500, or 2.7 percent of children under two who did not receive BF.

In the first case a percentage is given for EBF and in the second any breastfeeding; difficult to compare

Line 46 - I suggest to write "are exclusively breastfed" instead of receive EBF

Suggest to review the following - lines 48-49: In spite of recent increases in BF rates, Mexico remains far from achieving worldwide recommendations to protect, promote, and encourage the breastfeeding  The recommendations are for breastfeeding practices. and are well known interventions/actions to protect, promote and support/encourage breastfeeding (but not listed as worldwide recommendations)

Table 1. Education level. A number is missing under % of high school or more

As there were women with other children, it seems to me that previous breastfeeding experience may either help or interfere with intention to breastfeed. Was this explored?

Interesting the finding on sensitivity and specificity, is good that the authors commented on this.

Author Response

We appreciate the time and effort that you dedicated to providing feedback on our manuscript and are grateful for comments on and valuable improvements to our paper.

Please see below, in red, for a point-by-point response to your comments and concerns. All page numbers refer to the revised manuscript file with tracked changes.

Point 1: I suggest to review the following paragraph: the recommended breastfeeding rate is low worldwide.  In a recent study on the prevalence of extended breastfeeding in 94 low- and middle-income countries (LMICs), found that in 2018, roughly 55.2% of children under six months of age were not exclusively breastfed in the countries studied. In Mexico, the figure was 760,500, or 2.7 percent of children under two who did not receive BF. In the first case a percentage is given for EBF and in the second any breastfeeding; difficult to compare.

Response 1: We agree. We have changed it (see lines 38-43 in the revised manuscript):

The World Health Organization’s Global Nutrition Target (WHO GNT) proposes to achieve ≥70% LME compliance by 2030 [4]. Bhattacharjee et al.[5], in a recent study on the prevalence of exclusive breastfeeding in 94 low- and middle-income countries (LMICs), found that in 2018, roughly 55.2% of children under six months of age were not exclusively breastfed in the countries studied which represents a reduction of only 4.8% since the year 2000.

To facilitate a comparison between the international data cited above and the Mexican data, we have expressed all data in percentages (see lines 59-62 in the revised manuscript).

Before: “In Mexico, the figure was 760,500, or 2.7 percent of children under two who did not receive BF”

Now: “Fewer than a third of six-month-old children are exclusively breastfed (28.6%), while nearly half of children between 12 and 15 months (46.9%) and almost a third of children between 20 and 23 months (29%) continue to receive breast milk”

Point 2: Line 46 - I suggest to write "are exclusively breastfed" instead of receive EBF

Response 2: We appreciate your advice. In our earlier manuscript, we used “receive EBF” and we have changed to "are exclusively breastfed" as your recommendation. (see line 60  in the revised manuscript).

Point 3: Suggest to review the following - lines 48-49: In spite of recent increases in BF rates, Mexico remains far from achieving worldwide recommendations to protect, promote, and encourage the breastfeeding.  The recommendations are for breastfeeding practices. and are well known interventions/actions to protect, promote and support/encourage breastfeeding (but not listed as worldwide recommendations)

Response 3: We accept the correction; the recommendations have been established by the World Health Organization, which does not mean that they are applied and/or implemented worldwide. We modify the text as follows (lines 62-65 in the revised manuscript):

 In spite of recent increases in BF rates, Mexico remains far from achieving the WHO recommendations for breastfeeding practices [8], so to protect, promote, and encourage breastfeeding, research, and predictor identification it is so relevant.

Point 4: Table 1. Education level. A number is missing under % of high school or more

Response 4: Yes. Our mistake.  The omission was rectified.

Point 5: As there were women with other children, it seems to me that previous breastfeeding experience may either help or interfere with intention to breastfeed. Was this explored?

Response 5: Yes. We are aware that previous breastfeeding experience may either help or interfere with the intention to breastfeed; however, we decided to keep only the parity variable since it is a proxy variable of previous experience and better represented the sample studied considering the following:

  1. In our study, 55.3% of the 359 multiparous women included in the final sample reported prior experience with breastfeeding
  2. We found a positive association between the intention to breastfeed and prior experience with breastfeeding
  3. The analysis excluded more than 40% of the participants who had not had another child (Table 1)

Reviewer 2 Report

The study is very good and interesting for the reader. Breastfeeding is an important part of the reproductive process and due to many reasons this part , breastfeeding , is neglected by very different factors in many countries. I am impressed by this paper and it can be published in my view as it is. I have however two small  remarks , I don’t understand two sentences.

Line 40 and 41 I don’t understand this sentence. Does it mean that 2.7 percent of all children never receive breastfeeding??

And line 45, 46 , 47  is it so that the children continue to be breastfed.  The part of  the  sentence “continue to  breastfeed” I don’t understand . The word breastfeed means to give breastmilk not to receive it.

Otherwise I can agree with this paper and it can be accepted.

Author Response

We appreciate the time and effort that you dedicated to providing feedback on our manuscript and are grateful for comments on and valuable improvements to our paper.

Please see below, in red, for a point-by-point response to your comments and concerns. All page numbers refer to the revised manuscript file with tracked changes.

Point 1: Line 40 and 41 I don’t understand this sentence. Does it mean that 2.7 percent of all children never receive breastfeeding??

Response 1: In response to your suggestion, the last sentence is excluded and the data for Mexico are specified. We have changed it (see lines 38-43 and 59-62 in the revised manuscript).

The World Health Organization’s Global Nutrition Target (WHO GNT) proposes to achieve ≥70% LME compliance by 2030 [4]. Bhattacharjee et al.[5], in a recent study on the prevalence of exclusive breastfeeding in 94 low- and middle-income countries (LMICs), found that in 2018, roughly 55.2% of children under six months of age were not exclusively breastfed in the countries studied which represents a reduction of only 4.8% since the year 2000.

To facilitate a comparison between the international data cited above and the Mexican data, we have expressed all data in percentages.

Before: “In Mexico, the figure was 760,500, or 2.7 percent of children under two who did not receive BF”

Now (see lines 59-62 in the revised manuscript):Fewer than a third of six-month-old children are exclusively breastfed (28.6%), while nearly half of children between 12 and 15 months (46.9%) and almost a third of children between 20 and 23 months (29%) continue to receive breast milk”

Point 2: line 45, 46, 47 is it so that the children continue to be breastfed.  The part of  the  sentence “continue to  breastfeed” I don’t understand. The word breastfeed means to give breastmilk not to receive it.

Response 2: We accept the correction; we modify the text as follows (lines 59-60 in the revised manuscript):

Fewer than a third of six-month-old children are exclusively breastfed (28.6%)”.